# Cationic Copolymerization of Isobutylene with 4-Vinylbenzenecyclobutylene: Characteristics and Mechanisms

**DOI:** 10.3390/polym12010201

**Published:** 2020-01-13

**Authors:** Zhifei Chen, Shuxin Li, Yuwei Shang, Shan Huang, Kangda Wu, Wenli Guo, Yibo Wu

**Affiliations:** 1Beijing Key Lab of Special Elastomeric Composite Materials, Department of Materials Science and Engineering, Beijing Institute of Petrochemical Technology, Beijing 102617, China; chenzhifei1615@163.com (Z.C.);; 2College of Material Science and Engineering, Beijing University of Chemical Technology, Beijing 100029, China

**Keywords:** cationic polymerization, 4-vinylbenzocyclobutene, random copolymer, crosslink

## Abstract

A random copolymer of isobutylene (IB) and 4-vinylbenzenecyclobutylene (4-VBCB) was synthesized by cationic polymerization at −80 °C using 2-chloro-2,4,4-trimethylpentane (TMPCl) as initiator. The laws of copolymerization were investigated by changing the feed quantities of 4-VBCB. The molecular weight of the copolymer decreased, and its molecular weight distribution (MWD) increased with increasing 4-VBCB content. We proposed a possible copolymerization mechanism behind the increase in the chain transfer reaction to 4-VBCB with increasing of feed quantities of 4-VBCB. The thermal properties of the copolymers were studied by solid-phase heating and crosslinking. After crosslinking, the decomposition and glass transition temperatures (*T*_g_) of the copolymer increased, the network structure that formed did not break when reheated, and the mechanical properties remarkably improved.

## 1. Introduction

Over the years, isobutylene (IB) has played a crucial role in cationic polymerization due to its special structure. The living polymerization of IB has been realized, and researchers have also proposed living-polymerization mechanisms [1,2,3,4,5,6,7,8,9,10,11]. Copolymers with different structures and functions are obtained by the copolymerization of IB and other monomers [12,13,14,15,16]. Obtaining a functionalized copolymer by introducing a specific monomer during IB polymerization is an important research direction. Benzocyclobutene (BCB) and its derivatives are attracting considerable attention because of their special thermal properties. As shown in Scheme 1, the BCB’s strained four-membered ring opens when heated to >180 °C to form a high-activity intermediate, *o*-quinodimethane. This intermediate can react with dienophiles through the Diels–Alder reaction, thereby forming a six-membered ring. In the absence of dienophile monomers, the intermediate reacts with itself to form an eight-membered ring or a homopolymer of BCB [17,18,19,20].

Similar to styrene, 4-VBCB is cationically polymerizable. The electronegativity of the aromatic ring adjacent to the vinyl group of 4-VBCB stabilizes the vinyl secondary carbon to form a stable carbocation. Therefore, 4-VBCB can be introduced into cationic copolymerization with IB. The vulcanization of the copolymer of IB and 4-VBCB has significant advantages over the vulcanization of other IB-based copolymers, such as IB–isoprene rubber and brominated IB-*co*-p-methylstyrene [14]. The crosslinking of 4-VBCB involves only heat and does not need the addition of a vulcanizing agent and an accelerator, such as sulfur, zinc oxide, xanthates, and quinoid chemical reagents, which are harmful to the human body and difficult to remove from the rubber matrix completely [21]. Therefore, the copolymer of IB and 4-VBCB does not require the removal of small-molecule chemicals after crosslinking, and can be autoclaved without considering the volatilization of toxic substances, which is a crucial characteristic of biomedical materials.

Many studies have been conducted on the excellent properties of BCB-functionalized copolymers. The thermal and mechanical properties of BCB functional polymers and copolymers are drastically improved, and the dielectric constant is reduced after the thermal crosslinking reaction of the BCB ring. Therefore, the development of microelectronic materials exhibits great potential. Yang et al. synthesized oligomer poly(DVS-BCB-*co*-POSS) through the Heck reaction using 1,1,3,3-tetramethyl-1,3-divinyldisiloxane (DVS), BCB, and octavinyl-T8-silsesquioxane (vinyl-POSS). This oligomer has good thermal stability and low dielectric constants [22]. So et al. synthesized a copolymer of styrene and 4-VBCB through free radical polymerization and found that BCB dramatically increases the *T*_g_ of polystyene after ring opening and crosslinking [23]. Huang et al. synthesized a copolymer containing BCB and silacyclobutane side groups with low dielectric constants (approximately 2.30 MHz to 10 MHz) through atom transfer radical copolymerization [24]. The thermal crosslinking of BCB can also be applied in the preparation of nanoparticles. Sakellariou heated a poly(4-VBCB-*b*-butadiene) diblock copolymer in decane to induce the crosslinking of poly(4-VBCB) segments in the micelle core and obtain surface-functionalized nanospheres [25]. Harth prepared single-molecule nanoparticles by regulating the collapse of linear polymer chains [26]. BCB-functionalized polymers and copolymers are synthesized through free-radical [27,28] and anion polymerization [25,29]. Few works have investigated the cationic polymerization of 4-VBCB, except for the research of Sheriff et al. on the application of poly(styrene-*coblock*-4-VBCB)-polyisobutylene-poly(styrene-*coblock*-4-VBCB) in prosthetic heart valves [30]. Moreover, the mechanism underlying the cationic copolymerization of IB with 4-VBCB is unclear.

In reference to these results, we described the cationic random polymerization of IB and 4-VBCB with 2-chloro-2,4,4-trimethylpentane (TMPCl) as the initiator and TiCl_4_ as the coinitiator. Copolymerization laws were studied by changing the feed ratio of 4-VBCB. We proposed a possible polymerization mechanism using Gaussian 09 software for calculations. We also studied thermal and mechanical properties through the solid-phase crosslinking of poly(IB-*co*-4-VBCB).

## 2. Materials and Methods

### 2.1. Materials

IB (purity: 99.9%, Beijing Yanshan Petrochemical Co., Ltd., Beijing, China) and chloromethane (CH_3_Cl, purity: 99.9%, Beijing Yanshan Petrochemical Co., Ltd.) were dried in the gaseous state via passage through in-line gas-purifier columns packed with CaSO_4_ (purity: 99%, Beijing Chemical Reagents Company, Beijing, China)/drierite. They were condensed in a cold bath in a glove box prior to polymerization. N-hexane (purity: 99.5%, Beijing Chemical Reagents Company) was refluxed with sodium for several hours and distilled before use under nitrogen (N_2_) atmosphere. Dichloromethane (CH_2_Cl_2_, purity: 99.5%, Beijing Chemical Reagents Company) was dried over CaH_2_ (purity: 99.5%, Beijing Chemical Reagents Company) for one week and then rectified by reflux under N_2_ atmosphere. TMPCl was prepared by passing dry HCl into a 2,4,4-trimethyl-1-pentene (purity: 98.0%, Tokyo Chemical Industry Co., Ltd., Tokyo, Japan)/CH_2_Cl_2_ 50/50(*v/v*) mixture at 0 °C for 5 h; subsequently, the mixture was washed with NaHCO_3_ (purity: 99%, Beijing Chemical Reagents Company) until neutral, MgSO_4_ (purity: 99%, Beijing Chemical Reagents Company) was added to remove water, the mixture was filtered, and the filtrate was purified by using a rotary evaporator [31]. Titanium tetrachloride (TiCl_4_, purity: 99.0%, Energy Chemical, Shanghai, China) was purified from P_4_O_10_ (purity: 99.0%, Beijing Chemical Reagents Company) through distillation. 4-VBCB (purity: 97.5%, Bide Pharmatech Ltd., Shanghai, China), 2,6-di-tert-butylpyridine (DTBP, purity: 99.80%, Bide Pharmatech Ltd.), isopropanol (A.R. Beijing Chemical Reagents Company), and methanol (A.R. Beijing Chemical Reagents Company) were used as received.

### 2.2. Polymerization

The cationic copolymerization of IB and 4-VBCB was conducted in a glove box under dry N_2_ atmosphere. In a typical reaction, 15 mL of cyclohexane, 10 mL of CH_3_Cl, 3.54 × 10^−5^ mol TMPCl, 8.84 × 10^−5^ mol DTBP, and 1.13 × 10^−3^ mol TiCl_4_ were added sequentially to a screw-cap vial that was drained of water at −80 °C. The mixture was stirred evenly and held for 30 min to form TMPCl and TiCl_4_ complex. A mixture of 2 mL of IB (1.42 g, 0.025 mol) and 2.53 × 10^−3^ mol 4-VBCB dissolved in cyclohexane was added to the reaction mixture. Polymerization was continued for 2 h, and excessive methanol was added to terminate the reaction. Polymerization was quenched with 10 mL of prechilled ethanol. Polymer products were dried to a constant weight in a vacuum oven at 40 °C overnight. Monomer conversion was determined gravimetrically.

### 2.3. Measurements

Gel-permeation chromatography (GPC) was used to determine the molecular weights and molecular weight distributions (*M*_w_/*M*_n_) of the polymer. The unit was equipped with a Waters e2695 Separations Module, a Waters 2414 RI Detector, a Waters 2489 UV Detector, and four Waters styragel columns (E 2695, Waters, Milford, MA, USA) connected in the series of 500, 10^3^, 10^4^, and 10^5^ at 30 °C. Tetrahydrofuran was used as the mobile phase at a flow rate of 1 mL·min^−1^ at room temperature. PSS WinGPC software (Waters) was used to acquire and analyze chromatograms. The molecular weights of polymers were calculated relative to those of linear polystyrene standards. The microstructure of the polymer was analyzed by ^1^H NMR and ^1^C NMR using a nuclear magnetic resonance spectrometer (AVANCE AV400, Bruker, Switzerland). Tetramethylsilane was the internal standard and CDCl_3_ was the solvent. The polymers were subjected to thermogravimetric analysis using a thermogravimetric analyzer (TGA Q500, TA Instruments, New castle, DE, USA) in N_2_ atmosphere at 10 °C·min^−1^. The test temperature was 25 °C to 550 °C, and the heating rate was 10 °C·min^−1^. The glass transition temperature was analyzed using a differential thermal analyzer (DSC Q2000, TA Instruments) in a N_2_ atmosphere of 50 mL·min^−1^ gas flow. The test temperature was −90 °C to 150 °C, the temperature rise rate was 10 °C·min^−1^. The changes of crosslinked structure were characterized using a Fourier transform infrared spectrometer (Nicolet6700, Thermo Fisher Scientific, Waltham, MA, USA). The mechanical properties of the copolymer were tested by a microcomputer controlled electronic universal testing machine (Instron3366, Instron, Shanghai, China). The stretch rate was 40 mm·min^−1^.

## 3. Results and Discussion

### 3.1. Random Copolymers of IB and 4-VBCB

#### 3.1.1. Copolymerization of IB and 4-VBCB

Scheme 2 outlines the synthesis route of the poly(IB-*co*-4-VBCB) random copolymer. We first added IB to the initiation system to avoid the self-polymerization of 4-VBCB due to its high reactivity ratio. After the initiation of IB, a mixture of 4-VBCB and IB formulated at a certain ratio was gradually added to the reaction. Polymerization was continued for 2 h. Finally, excessive prechilled methanol was used to terminate the reaction.

The structures of the copolymer were characterized by ^1^H NMR. Figure 1a shows the ^1^H NMR spectrum of polyisobutylene (PIB). The characteristic resonance values at δ = 1.1 ppm (peak a) and δ = 1.4 ppm (peak b) were assigned to methyl and methylene, respectively. Figure 1b shows the ^1^H NMR spectrum of 4-VBCB. Aromatic ring proton peaks were located at δ: 6.6–7.24 ppm (peak c), vinyl proton peaks were located at δ = 5.1 ppm (peak a) and δ = 5.7ppm (peak b), and two methylene protons peak of BCB four-member ring were found at δ = 3.1 ppm (peak d). Figure 1c shows the ^1^H NMR spectrum of the poly(IB-*co*-4VBCB) random copolymer, and the characteristic peaks of each proton in poly(IB-*co*-4VBCB) random copolymer were consistent with the structures. Comparing Figure 1c with Figure 1a,b reveals that the ^1^H NMR spectrum of the random copolymers lacked vinyl protons at δ = 5.1 ppm and δ = 5.7 ppm, and protons associated with the P(IB) and P(4-VBCB) segment backbone and BCB side groups were present. The poly(IB-*co*-4VBCB) random copolymer was successfully synthesized.

#### 3.1.2. Reactivity Ratio of IB and 4-VBCB

The reactivity of monomers directly determines the copolymer composition. Thus, we first studied and calculated the reactivity ratio of IB and 4-VBCB using the Yezreielv–Brokhina–Roskin method [32]. We obtained a series of copolymers with different monomer compositions by changing the feed ratio of 4-VBCB to IB. The polymerization lasted for only 20 min to ensure that conversion rates were less than 15%. Then, the content of two monomers in the copolymer was calculated on the basis of the integrated area of the corresponding peak in the ^1^H NMR spectra (Table 1). The calculation method is as follows: $$F_{\mathrm{IB}}=\frac{A_{1.1}}{6}/\left(\frac{A_{1.1}}{6}+\frac{A_{3.1}}{4}\right)$$
$$F_{4-VBCB}=\frac{A_{3.1}}{4}/\left(\frac{A_{1.1}}{6}+\frac{A_{3.1}}{4}\right)$$
where *A*_1.1_ is the resonance area of the two –CH_3_ on isobutene, and *A*_3.1_ is the resonance area of the four-membered ring on 4-VBCB.

In accordance with the Yezreielv–Brokhina–Roskin method, we calculated monomer reactivity with the following equation:$$\left(\frac{x}{y^{1/2}}\right)r_{1}-\left(\frac{y^{1/2}}{x}\right)r_{2}+\left(\frac{1}{y^{1/2}}-y^{1/2}\right)=0$$
where x is the starting molar ratio of IB to 4-VBCB; y is the molar ratio of IB to 4-VBCB in the copolymer; and r_1_ and r_2_ are the reactivity ratio of IB and 4-VBCB, respectively, which are calculated as follows: $$r_{1}=\left[\sum_{i=1}^{n}\frac{y_{i}}{x_{i}^{2}}\times \sum_{i=1}^{n}x_{i}\left(1-\frac{1}{y_{i}}\right)+n\sum_{i=1}^{n}\frac{y_{i}}{x_{i}}\left(\frac{1}{y_{i}}-1\right)\right]/\left(\sum_{i=1}^{n}\frac{x_{i}}{y_{i}^{2}}\times \sum_{i=1}^{n}\frac{y_{i}}{x_{i}^{2}}-n^{2}\right),$$
$$r_{2}=\left[\sum_{i=1}^{n}\frac{x_{i}^{2}}{y_{i}}\times \sum_{i=1}^{n}\frac{y_{i}}{x_{i}}\left(\frac{1}{y_{i}}-1\right)+n\sum_{i=1}^{n}x_{i}\left(1-\frac{1}{y_{i}}\right)\right]/\left(\sum_{i=1}^{n}\frac{x_{i}}{y_{i}^{2}}\times \sum_{i=1}^{n}\frac{y_{i}}{x_{i}^{2}}-n^{2}\right).$$

The reactivity ratios of IB and 4-VBCB are *r*_1_ = 0.47 and *r*_2_ = 2.08, respectively. Although 4-VBCB tended to self-polymerize, the two monomers tended to form random copolymers when its concentration was considerably lower than IB.

#### 3.1.3. Characteristics and Mechanisms

The controlled polymerization of IB initiated by TMPCl/TiCl_4_/DTBP system was achieved (Appendix A). To verify the living characteristics of the random copolymerization of 4-VBCB and IB, we conducted a series of experiments where only the 4-VBCB feed ratio was changed, and other conditions remained unchanged. The details are shown in Table 2. Figure 2 shows the GPC traces of PIB segments and random copolymers obtained with different 4-VBCB feeding quantities. As the feed quantities of 4-VBCB were increased, the peak position of the random copolymers moved toward low molecular weights, and the widths of the peak gradually increased. Figure 3 shows the average molecular weight *M_n_* and the MWD of the PIB and the random copolymers obtained from various experiments as a function of the feeding quantity of 4-VBCB. The *M_n_* value gradually decreased with the increase in 4-VBCB feed quantity, whereas MWD exhibited an increasing trend.

The most likely reason for this result was that the chain transfer reaction to 4-VBCB increased rapidly with increasing feeding quantities of 4-VBCB. The active center of the copolymer chain was more likely to transfer to 4-VBCB with the simultaneous existence of two monomers. The copolymers of IB and 4-VBCB were assumed to have only four chain end groups, α-olefin, β-olefin, *tert*-Cl [33], and indane-BCB. Figure 4 shows the ^1^H NMR spectra of the four end groups. The fractional molar amount of each chain end was calculated separately as follows:$$F_{\alpha -\mathrm{olefin}}=\frac{\left(A_{4.64}+A_{4.85}\right)/2}{\left(A_{4.64}+A_{4.85}\right)/2+A_{1.68}/6+A_{1.7}/6+\left(A_{2.51}+A_{2.83}\right)/2}$$
$$F_{\beta -\mathrm{olefin}}=\frac{A_{1.7}/6}{\left(A_{4.64}+A_{4.85}\right)/2+A_{1.68}/6+A_{1.7}/6+\left(A_{2.51}+A_{2.83}\right)/2}$$
$$F_{tert-Cl}=\frac{A_{1.68}/6}{\left(A_{4.64}+A_{4.85}\right)/2+A_{1.68}/6+A_{1.7}/6+\left(A_{2.51}+A_{2.83}\right)/2}$$
$$F_{indane-BCB}=\frac{\left(A_{2.51}+A_{2.83}\right)/2}{\left(A_{4.64}+A_{4.85}\right)/2+A_{1.68}/6+A_{1.7}/6+\left(A_{2.51}+A_{2.83}\right)/2}$$
where *A*_4.64_ and *A*_4.85_ are the resonance areas of *α*-olefin, *A*_1.68_ is the resonance area of the two methyl groups attached to β-olefin, and *A*_1.7_ is the area of two methyl groups resonance attached to the *tert*-*Cl* carbon. *A*_2.51_ and *A*_2.83_ are related to the methylene protons of the indane structure produced by chain-transfer to 4-VBCB. The results are shown in Table 3.

The corresponding reaction mechanism of IB and 4-VBCB cationic copolymerization initiated by TMPCl/DTBP/TiCl_4_ was proposed on the basis of the results for the end groups in the above copolymerization reaction (Scheme 3). In the initiation reaction, TiCl_4_ extracted –Cl from TMPCl to induce the generation of carbon cationic and formed the [TiCl_5_]^−^ counterion. With the help of the proton trapping agent DTBP, electrophilic carbocation preferentially attacked the monomers IB and 4-VBCB to initiate polymerization. Various chain transitions and chain terminations occurred frequently during polymerization. Chain transfer broke the chain by transferring the active center to the monomers IB and 4-VBCB. The broken polymer chain generated two groups at the end, namely, α-olefin and β-olefin, when the reaction chain was transferred to IB and the indane-BCB end group when the reaction chain was transferred to 4-VBCB. Then, the new carbocation active center restarted the reaction to produce a new polymer chain. Moreover, the growing carbocation combined with the negative part of the counter ion to terminate the chain and produce the *tert*-Cl end group.

### 3.2. Thermal Properties of Poly(IB-co-4-VBCB) Random Copolymer

Figure 5 shows the DSC curve of the poly(IB-*co*-4-VBCB) containing 6 mol % of 4-VBCB. It shows the heat change of the copolymer with temperature raised from −90 °C to 350 °C, retained for 10 min at 260 °C for crosslinking, cooled to −90 °C, and reheated to 350 °C. *T*_g_ was 25.76 °C during the first heating process. The BCB ring began to open when the copolymer was heated to 225 °C, and the maximum BCB ring-opening temperature was approximately 257 °C. *T*_g_ was 38.56 °C during the second heating process after BCB ring opening, which was higher than the first *T*_g_. This finding could be attributed the crosslinking of the BCB ring. The new C–C bond formed by crosslinking was stable and did not open even when the copolymer was heated to 350 °C.

We studied the ^1^H NMR spectrum of the copolymer to further determine the crosslinking structure of the copolymer. As shown in Figure 6, the absence of a signal peak (peak a) at 3.11 ppm on the ^1^H NMR spectrum of the crosslinked copolymer, indicated that all of the BCB olefin rings were open. A crosslinked eight-membered ring proton peak can be found at 3.18 ppm (peak b). The new signal peak at 5.15 ppm (peak c) may be ascribed to the structure in which no crosslinking occurred after the opening of the BCB ring. In the ^13^C NMR spectra (Figure 7), the characteristic signals of the PIB segment appeared at 31.4, 38.3, and 59.7 ppm, and the characteristic signals of the P4VBCB segment appeared at approximately 29.7 ppm. After crosslinking, we found that the characteristic signals (peak f) of the BCB four-member ring disappeared in the ^13^C NMR spectra, and a new signal peak (peak d), which was attributed to the crosslinking structure appeared. The same evidence can be observed from the FTIR spectrum of the copolymer (Figure 8). The bands at 2949, 1471, 1388, 1365, and 1230 cm^−1^ were attributed to the PIB segment. The bands at 823 and 710.05 cm^−1^ disappeared after crosslinking, whereas a new absorption peak appeared at 1072 cm^−1^, thereby corresponding to the opening of the four-membered ring of BCB and the formation of the eight-membered ring structure by crosslinking.

The thermal stability of the copolymer containing 6 mol % 4-VBCB is shown in Figure 9. The epitaxial starting temperatures of the uncrosslinked and crosslinked copolymers were 390 °C and 406 °C, respectively. This result was due to the increased thermal stability caused by the crosslinking of the copolymer.

### 3.3. Mechanical Properties of Poly(IB-co-4-VBCB) Random Copolymer.

We studied the mechanical properties of copolymers after thermal crosslinking. The copolymer with a molecular weight of approximately 40,000 was placed in a dumbbell mold. Then, the copolymer was heated to 250 °C at 4 MPa for 10 min. During heating, the copolymer became viscous at 250 °C. The 4-VBCB’s four-membered ring opened and reacted with itself to form solid C–C bond. The stress-strain behavior of the module is illustrated in Table 4 and Figure 10. As the components of 4-VBCB in the copolymer increased, the elongation at break of the crosslinked copolymer increased slightly, and the maximum tensile stress did not change considerably. The mechanical properties of the copolymer were significantly improved compared with its viscous and gummy form before crosslinking.

## 4. Conclusions

A poly(IB-*co*-4-VBCB) random copolymer was successfully synthesized through cationic polymerization initiated by the TMPCl/DTBP/TiCl_4_ system. The reactivity ratios of IB and 4-VBCB, which were r_1_ = 0.47 and r_2_ = 2.08, respectively, in the copolymerization reaction were calculated using the Yezreielv–Brokhina–Roskin method. The two monomers formed a random copolymer. The molecular weight and molecular weight distribution of the copolymer were characterized by GPC. The content of 4-VBCB in the copolymer also increased with the increase in the feed ratio of 4-VBCB to IB. However, the molecular weight of the copolymer decreased while the molecular weight distribution increased. The possible reaction mechanism was that the chain transfer reaction to 4-VBCB increased with the increase in the feed ratio of 4-VBCB to IB, thereby breaking the polymer chains. The thermal properties of the copolymer were determined by DSC and TGA. When the copolymer was heated to approximately 220 °C, the BCB ring began to open, and the newly formed intermediates crosslinked with each other to form a network structure between copolymer chains. The crosslinked structure did not break even when the copolymer was reheated to the temperature at which the BCB ring can be opened. At the same time, the glass transition and decomposition temperatures of the copolymer increased after crosslinking due to its thermal crosslinking property. The state of the copolymer changed from viscous to fixed with a certain mechanical strength after crosslinking.

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
