# Peer review of "Cationic Copolymerization of Isobutylene with 4-Vinylbenzenecyclobutylene: Characteristics and Mechanisms"

_polymers, 2020, doi:10.3390/polym12010201_

Round 1
Reviewer 1 Report
The manuscript reports on the copolymers prepared by a cationic copolymerization of isobutene with 4-vinyl-benzene-cyclobutilene initiated by a combination of 2-cloro-2,4,4-trimethylpentane and TiCl4.
There is plenty of drawbacks and flaws, which must be revised, such as :
1) There is a complete absence of any 13C NMR data for copolymers isolated from polymerization experiments, and also for those that are believed to be cross-linked obtained by thermal treatment (e.g., during thermic analyses). Only a few obscure 1H NMR spectra are shown, which definitely cannot provide any conclusive information about the microstructure of copolymers. Also, it is absolutely not clear how the assignments of signals from cross-links and chain-ends have been done as no any reference to literature precedents is provided.
2) What is the regular range of polydispersity values (PDI) experimentally obtained in a classical “living” polymerization process ? Probably, 1.01-1.20 ? If the range of experimental PDI values is superior to 1.20 (but still below 2.0 ) the polymerization has a “living” character, but is necessarily contaminated by side processes like termination or irreversible chain transfer.
In this study, the polymerization is not “living”, at least, under actual experimental conditions.
3) The energy values obtained from computations have very poor if any physical sense, since the models of species used for calculation are unreal. Then, in terms of the calculated energy values, paths c, e, e and f will never be realized.
4) Excepting the reaction with 4-VBCB, the mechanism shown on Scheme 3, which is introduced as it was deduced from experiment, is basically the same given for cationic polymerization of vinyl monomers in school books. At the same time, the quality of the experimental data does not allow exact identification of the nature of cross-links.
5) Looking at Figure 6, whether should we think that the cross-linking process is endothermic by nature ? If so, then why ? Also, the determination of the glass-transition point (Tg) values appears to be vague and uncertain. DMA would better to be used for this purpose.
6) lines 81-83: chloromethane (?!) was refluxed with sodium ?!
7) line 85: CaH2 should be instead of GaH2.
Reviewer 2 Report
Well written, a few mistakes, please make a general review. They’ve presented a complete amount of evidence of several experiments, included mechanical properties that is adequate for potencial applications.
Author Response
Thank you very much for the recognition of our work.
Reviewer 3 Report
This manuscript reports the cationic copolymerization of isobutylene with 4-2-vinylbenzenecyclobutylene. In addition to the characterization of the copolymers, mechanical tests and mechanistic investigations have also been performed.
It is an interesting manuscript, but it cannot be published in Polymers in its present form.
Comments:
The % yield of the copolymerization reaction is not reported.
Line 85: “GaH2” is not correct. Do the authors mean “CaH2”?
Lines 90-92: The sentence needs rephrasing. In addition, P2O5 is not the correct formula; the correct formula is P4O10.
Lines 93-97: I do not understand what the authors mean.
Line 138: How was the equation derived? The authors need to elaborate.
Line 140: r1 and r2 are the reactivity rates of IB. r1 and r2 are not rates. They are the reactivity ratios of IB and 4-VBCB, as the authors state in line 143.
Lines 141 and 142: How were the equations derived? How were the reactivity ratios calculated?
Line 146, Table 1: How was the composition of the copolymers calculated?
Line 154, Scheme 2: 4-VBCB should be added to the second step of the reaction.
Line 160: “and the characteristic resonances were found at δ =3.1 ppm”. Characteristic of what?
Lines 161-162: Figure 1(b) and Figure 1(b).
Line 224: “… to induce the carbon cationic and [TiCl5]- gegenion”. The sentence needs rephrasing.
Line 268, Figure 9: The y axis is not correct. The % weight cannot be higher than 100.
Line 269, Section 3.3: That section is not well-written. It is not clear how the experiments were performed. It is also not clear what happened to the polymer during the preparation of the sample.
Please check the text very carefully for spelling, grammar and punctuation errors.
Reviewer 4 Report
In this manuscript, Wu and coworkers demonstrated cationic copolymerization of 4-vinylbenzenecyclobutene (VBCB) and isobutylene (IB). The copolymerization laws and polymerization mechanisms were discerned via changing the feed ratio of two monomers as well as in silico modeling. Moreover, the authors found thermal property and mechanical property of crosslinked poly(VBCB-co-IB) outperformed their uncrosslinked analogues. This is a well-designed study and the data can support their conclusions and proposals. The writing is clear and allow authors to follow smoothly. Therefore, I would like to support publication of this manuscript in Polymers. But the authors should address my comments below:
What is the GPC standard used in this study? PMMA or PS?2. In the conclusion of this manuscript, the authors said "the molecular weight and molecular weight distribution of copolymer decreased". This is not fully correct. According to table 1, only molecular decreased. MWD increased as feed ratio of VBCB increased.
3. In figure 7, the label "c" should be assigned to the chemical structure. Also, fix the spelling typo in the caption. "spectr" to "spectra".
4. The mechanical properties in table 4 don't match with figure 10. For example, sample B has a maximum tensile stress at 0.53 MPa in table 4. However, this value is definitely below 0.5 MPa according to the stress-strain curve in figure 10. Please verify them.
5. Please check and remove the sentences on Page 3, lines 94-97. Those sentences are from the journal template.
Round 2
Reviewer 1 Report
The energy values obtained from computations have very poor if any physical sense, since the models of species used for calculation are irrelevant.
For example, how the ortho-deprotonation of aromatic ring in 4-VBCB can be possible if the calculated energy (enthalpy) for the corresponding reaction is > 700 kcal/mol ? Thus, in terms of the calculated energy values, paths c, e, e and f will never be realized.
Given B3LYP, used for these calculations, is exchange-correlation functional, and not exact exchange functional, it is not clear why and how the HF energy values were extracted.
Author Response
Point: The energy values obtained from computations have very poor if any physical sense, since the models of species used for calculation are irrelevant. For example, how the ortho-deprotonation of aromatic ring in 4-VBCB can be possible if the calculated energy (enthalpy) for the corresponding reaction is > 700 kcal/mol? Thus, in terms of the calculated energy values, paths c, e, e and f will never be realized. Given B3LYP, used for these calculations, is exchange-correlation functional, and not exact exchange functional, it is not clear why and how the HF energy values were extracted.
Response: We strongly agree with the reviewer's opinion that proton transfer with energy changes greater than 700 Kcal/mol is indeed impossible. We have adopted a more reasonable calculation method. Compared with B3LYP, M062X is more reliable in the calculation of chemical thermodynamics of charge transfer of main group elements, and the calculated energy change is less than 700Kcal/mol. We added the relevant calculation data in the supporting file. At the same time, the polymerization mechanism was adjusted accordingly. In the previously proposed mechanism, we considered the loss of H+ in the 4-VBCB active center at the end of the polymer chain to form a + 2-valent cationic, which is obviously wrong. The calculated energy is also very large. In the newly proposed mechanism, we consider that a 4-VBCB near the end of the chain loses H+, and then the 4-VBCB at the end of the chain is coupled with it to form indene structure. So, the calculated energy is much less (page 7, line 199; page 8, lines 209-210; page 10, lines 239-240).
Reviewer 3 Report
The authors have addressed my previous comments satisfactorily.
The text still needs close and careful reading and editing.
Author Response
Point: The authors have addressed my previous comments satisfactorily. The text still needs close and careful reading and editing.
Response: The English of the whole manuscript was smoothed and revised again. We wish it could be understood easier.
Round 3
Reviewer 1 Report
I would suggest removing the computational part, since it doesn't deal with the polymerization mechanism, rather it treats a few reactions of proton (H+) addition onto vinyl bonds, which are only poorly representative of the possible reactivity pattern.
Author Response
Comment: I would suggest removing the computational part, since it doesn't deal with the polymerization mechanism, rather it treats a few reactions of proton (H+) addition onto vinyl bonds, which are only poorly representative of the possible reactivity pattern.
Response: We agree with the reviewer's opinion on simulation calculations, and we have removed the computational part of the manuscript.